# Care-experienced cHildren and young people's Interventions to improve Mental health and wEll-being outcomes: Systematic review (CHIMES) protocol

Rhiannon Evans [1] Maria Boffey,[2] Sarah MacDonald,[1] Jane Noyes [3] G.J. Melendez-Torres,[4] Helen E Morgan [5] Rob Trubey [6] Michael Robling [1,6] Simone Willis [5] Charlotte Wooders[2]

[1]DECIPHer, School of Social Sciences, Cardiff University, Cardiff, UK
[2]The Fostering Network in Wales, Cardiff, UK
[3]School of Health Sciences, Bangor University, Bangor, UK
[4]Peninsula Technology Assessment Group (PenTAG), University of Exeter, Exeter, UK
[5]Specialist Unit for Review Evidence, Cardiff University, Cardiff, UK
[6]Centre for Trials Research, Cardiff University, Cardiff, UK

**Correspondence to**
Dr Rhiannon Evans;
evansre8@cardiff.ac.uk

## ABSTRACT

**Introduction** The mental health and well-being of children and young people who have been in care (ie, care-experienced) are a priority. There are a range of interventions aimed at addressing these outcomes, but the international evidence-base remains ambiguous. There is a paucity of methodologically robust systematic reviews of intervention effectiveness, with few considering the contextual conditions under which evaluations were conducted. This is important in understanding the potential transferability of the evidence-base across contexts. The present systematic review will adopt a complex systems perspective to synthesise evidence reporting evaluations of mental health and well-being interventions for care-experienced children and young people. It will address impact, equity, cost-effectiveness, context, implementation and acceptability. Stakeholder consultation will prioritise a programme theory, and associated intervention, that may progress to further development and evaluation in the UK.

**Methods and analysis** We will search 16 bibliographic databases from 1990 to June 2020. Supplementary searching will include citation tracking, author recommendation, and identification of evidence clusters relevant to included evaluations. The eligible population is children and young people (aged ≤25 years) with experience of being in care. Outcomes are (1) mental, behavioural or neurodevelopmental disorders; (2) subjective well-being; (3) self-harm; suicidal ideation; suicide. Study quality will be appraised with methodologically appropriate tools. We will construct a taxonomy of programme theories and intervention types. Thematic synthesis will be used for qualitative data reporting context, implementation and acceptability. If appropriate, meta-analysis will be conducted with outcome and economic data. Convergent synthesis will be used to integrate syntheses of qualitative and quantitative data.

**Ethics and dissemination** We have a comprehensive strategy for engagement with care-experienced children and young people, carers and social care professionals. Dissemination will include academic and non-academic publications and conference presentations. Ethical approval from Cardiff University's School of Social Sciences REC will be obtained if necessary.

**PROSPERO registration number** CRD42020177478.

## Strengths and limitations of this study

► This review will offer one of the first methodologically robust syntheses of evidence reporting evaluations of mental health and well-being interventions for care-experienced children and young people.

► In taking a complex systems approach, the review will provide insights into the potential transferability of interventions and the replicability of effects across contexts.

► From this review, the study team aim to identify a potential programme theory (and associated intervention) that can progress to further development and evaluation in the UK context.

► The review will support policy-makers and practitioners in decision-making about how to support the mental health and well-being of children and young people who have been in care.

► The review is limited by only including studies published in higher resource countries.

## BACKGROUND
### Rationale
The mental health and well-being of children and young people who have been in care (ie, care-experienced) is a growing public health and social care concern. Internationally, almost 50% of individuals involved in the child welfare system have a diagnosable mental health condition.[1] Meanwhile in the UK, they are nearly five times as likely to have at least one psychiatric diagnosis compared with the general population.[2] Care-experienced individuals are at an elevated risk of poor subjective well-being[3] and are more than four times as likely to attempt suicide .[3] Adverse mental health outcomes incur significant health and social care costs, often due to the associated risk of placement instability and breakdown.[5 6] Evidence from the UK reports that insecure and unstable care placements cost £22 415 more per child per annum (across

health, social care and criminal justice) than stable care pathways.[7]

There is a clear need to improve mental health provision for those who experience care, with NICE guidance stating that the UK evidence base does not adequately serve this population.[8] Equally, while the Department for Education and Department of Health and Social Care's joint statutory guidance has mandated that local authorities ensure timely and adequate services,[9] there are reported incidents of failure to identify need, overly stringent eligibility thresholds, and withholding of care where there is not a stable placement.[10] A comprehensive synthesis is required to identify evidence-based interventions in order to develop recommendations to enhance policy and practice.

Internationally there is a large evidence base for interventions,[11–22] although in the UK it remains emergent. Current interventions focus on preventative approaches that operate across socioecological domains, including community level change (eg, Flying Start),[23] functioning within the care placement,[24 25] the theoretical orientation of social care teams (eg, adoption of a trauma informed model),[26] and the social and emotional competencies of the individual child or young person.[27] Meanwhile, treatment has tended to focus on both the type and availability of therapeutic approaches, such as counselling or access to mental health services.[28]

Despite the size of the extant literature, there are limitations that need to be attended to, notably in regard to existing systematic reviews. First, the most inclusive syntheses lack rigorous methodologies. One of the most recent and relevant literature reviews recognised that it is not exhaustive.[11] Second, reviews tend to focus on a limited number of intervention types (eg, treatment foster care),[12] populations (eg, foster care),[12 14 18 27 28] or outcomes (eg, externalising behaviours).[18 19] This makes it difficult to know if a particular intervention can work effectively with different populations across multiple outcomes in order to reduce the number of approaches needing to be resourced within a system. Third, reviews often include only rudimentary treatment of context and process data. Understanding the complex system in which interventions operate is imperative. International variations in social care and healthcare systems may constrain the transferability of evidence and the extent to which we can 'borrow strength' from the original evaluation.[29] For example, Multidimensional Treatment Foster Care (MTFC) is shown to not be effective in Sweden, despite demonstrating effects in the USA, as it is essentially equal to usual care.[30]

The present mixed method systematic review will use a robust methodology to draw together the evidence-base for interventions aimed at improving the mental health and well-being of care-experienced children and young people, including those currently in care and those who have previously been in care. Adopting a complex systems perspective,[31 32] we will synthesise evidence of impact, equity and cost effectiveness, while also focusing on understanding how intervention effects are contingent on the context in which they are evaluated. In collaboration with stakeholders (young people, carers and social care professionals), the resulting synthesis will be used to identify, where possible, an overarching programme theory (or theories) that may inform the development or adaptation of an intervention to progress to further evaluation within the UK context.

## Review aims and questions

The study aims to systematically review and synthesise international evidence on interventions to improve the mental health and well-being of care-experienced children and young people. We will address the following research questions:

RQ1: What are the types, theories and outcomes tested in mental health and well-being interventions for care-experienced children and young people?

RQ2: What are the effects (including inequities and harms) and economic effects of interventions?

RQ3: How do contextual characteristics shape implementation factors, and what are key enablers and inhibitors of implementation?

RQ4: What is the acceptability of interventions to key stakeholders?

RQ5: Can and how might intervention types, theories, components, and outcomes be related in an overarching system-based programme theory?

On the completion of the initial phases of the systematic review (RQ1–5), the study will undertake a second phase of research and address a further research question:

RQ6: Drawing on the findings from RQ1–5, what do stakeholders think is the most effective, feasible and acceptable intervention in the UK that could progress to further outcome or implementation evaluation?

## METHODS

### Protocol and registration

The protocol is registered with PROSPERO and reported in accordance with the Preferred Reporting Items for Systematic reviews and Meta-Analyses Protocols (PRISMA-P) guidelines.[33] The review will be published in accord with the PRISMA guidelines,[34] with qualitative synthesis being structured by the ENTREQ reporting standards[35] and the mixed methods process evaluation synthesis following the principles prescribed by Cochrane.[36]

### Eligibility criteria

The eligibility parameters are reported in accordance with the PICOS framework and presented in the logic model (online supplemental A).

*Population:* children and young people aged ≤25 years olds. Where the target population extends beyond the age of 25 years old (eg, 18–30 years old), subgroup data must be provided for participants aged ≤25 years olds. Currently placed in care or previous care experience. Period of time in care is not restricted. Care can include

in-home care (ie, voluntary transfer of parental responsibility to statutory services) and out-of-home care (eg, foster care, residential care, or formal kinship care). No restriction on baseline mental health conditions or uptake of pharmacological treatment. The following are excluded: general population; children in need; individuals at the edge of care; care without specified statutory involvement (eg, informal kinship care); adoption; and refugees. Any population may be targeted as intervention participants (eg, carer, birth family, teacher or social worker) but outcomes must be reported at the level of the child or young person.

*Intervention:* any attempt to disrupt existing system practices. They may be monocomponent or multicomponent and operate across any of the following socioecological domains: individual; interpersonal; organisational; community; and policy/legal. Interventions may focus on promotion/prevention or the management/treatment of symptomology. There are no a priori criteria for implementation (ie, delivery setting, delivery mode, delivery agent). The following are excluded: pharmacological interventions, either as a single component approach or multicomponent intervention (eg, combined with psychosocial activities).

*Comparator:* treatment as usual; other active treatment; no specified treatment.

*Outcomes:* three domains of primary outcomes: (1) mental, behavioural or neurodevelopmental disorders as specified by ICD-11; (2) Subjective well-being (eudaimonia and hedonia); (3) self-harm; suicidal ideation; suicide. Measures may use dichotomous, categorical or continuous variables. Outcomes may be ascertained through clinical assessment, self-report or report by another informant (eg, teacher).

*Study design: programme theory:* describe intended theory, logic model or mechanisms of effect. May include mediation analysis (RQ1); *outcome evaluation:* (individual/cluster) randomised controlled trials and quasiexperimental study designs. We exclude post measurement only or pre–post measurement in intervention group only (RQ1; RQ2); *process evaluation:* all qualitative and quantitative study designs (RQ1; RQ3; RQ4). Included studies must empirically report on relevant contextual influences, implementation or acceptability; *economic evaluation:* must relate costs to benefits and can report: cost-minimisation, cost-effectiveness, cost utility or cost–benefit analysis. May be model based or trial based. Decision-analytic models capturing intervention impacts on mental health and well-being will be eligible (RQ2). *Relationship between study designs:* to be included programme theory papers must have an associated empirical outcome evaluation. Process evaluations and economic evaluations do not necessarily have to be linked to an empirical outcome evaluation, as they provide wider contextual insight into how interventions interact with complex system characteristics.

## Information sources

The following electronic bibliographic databases will be searched: ASSIA, British Education Index, Child Development & Adolescent Studies, CINAHL, Embase, ERIC, Cochrane Database of Clinical Controlled Trials (CENTRAL), HMIC, IBSS, Medline (Medline in Process and Medline ePub), PsycINFO, Scopus, Social Policy & Practice, Social Services Abstracts, Sociological Abstracts and Web of Science (Social Sciences Citation Index, Conference Proceedings Citation Index—Social Science & Humanities, Emerging Sources Citation Index). Supplementary searching techniques will include citation tracking of included studies, contacting international experts, searching trial registers and consulting websites of key social and healthcare organisations. Searches will be conducted for 'clusters' of reports related to included studies to develop descriptions of programme theories and further progress understanding of context.[37 38] Inclusion will be restricted from 1990, which marked the ratification of the UN Convention on the Rights of the Child.[39] This prescribed comprehensive social and healthcare provision for children internationally and started the proliferation of intervention in this area. To maximise applicability of evidence to the UK, studies from low-income and middle-income countries will be excluded. All languages will be included in searches.

## Search strategy

We developed a provisional search strategy in Ovid Medline (online supplemental B). It will be adapted to the functionality of each database on study commencement.

## Data management and study selection

Data will be exported to endnote for deduplication and then to EPPI Reviewer Web for screening. In the first instance reference titles will be screened on the basis of title to identity clearly irrelevant retrievals (eg, pharmacological treatments), with these exclusions being verified by a second reviewer. Titles and abstracts of remaining studies will be screened independently and in duplicate. The inclusion criteria proforma will be piloted and calibrated by two reviewers screening the same 100 title and abstracts, with disagreement being resolved through consensus or recourse to a third reviewer. Full texts will be retrieved and appraised for study inclusion. Conflicts will be resolved through discussion or recourse to a third reviewer. A record of the selection process will be retained in adherence with the PRISMA flow diagram.

## Data collection

Once the final number of included studies is confirmed, we will chart clusters of studies, constructing a visual knowledge map similar to that recommended within scoping reviews.[40 41] This process will support decision-making about the depth of data extraction at the next stage (eg, sampling may be conducted for qualitative studies) and extent of synthesis to be undertaken (eg, meta-analysis may be conducted where there are multiple

studies). The cHildren and young people's Interventions to improve Mental health and wEll-being outcomes: Systematic review (CHIMES) review has a study advisory group that will be consulted at this stage to confirm the extraction and synthesis.

Standardised data extraction forms will be developed and calibrated with a subset of studies for input to EPPI Reviewer Web. Two versions of extraction forms may be developed based on the mapping of the data. The first will be for studies included in the review but not included in more in-depth synthesis, and the second for studies subjected to more in-depth synthesis. For primary qualitative studies a subset of data will be coded in vivo to develop a preliminary coding tree with a priori codes, but which can be amended to incorporate emergent codes. On confirmation of the extraction forms and coding tree, two reviewers will independently extract and code data from 10% of studies, with the remainder being extracted by one reviewer and verified by a second.

### Data items
For all studies data will be extracted on: study characteristics, participant demographics; setting; study design and methods; intervention theory of change and intervention characteristics. For outcome evaluations data will be extracted for: measurements, sequence generation, allocation concealment, blinding, data completeness, baseline differences and adjustment for difference, control of confounding, and outcomes at follow-up at both population and subgroup level. Data items from process evaluations will be the implementation theory, implementation strategy and implementation agents. Quantifiable assessments of implementation will be reach, receipt, and fidelity. Contextual characteristics will be classified according to the context and implementation of complex interventions (CICI) framework: geographical; epidemiological; legal; socioeconomic; sociocultural; ethical; and political.[42] Acceptability data across stakeholders will be extracted. Economic evaluation data items will be intervention and comparator costs, perspective, structural and empirical inputs, time horizon, and cost-effectiveness. Health utility data will also be extracted. Where there is incomplete data the author will be asked for additional information and where this is not retrieved the data will be reported as missing and included in the risk of bias assessment.

### Risk of bias (individual studies)
Study data will be appraised with a methodologically appropriate tool. Programme theory will be appraised using a tailored appraisal tool developed by members of the research team.[43] Domains of assessment will be: clarity of constructs; clarity of relationships between constructs; testability; parsimony and generalisability.

Outcome evaluations will be assessed with the relevant tool prescribed by the Cochrane Handbook for Systematic Reviews of Interventions.[44] In the case of randomised controlled trials, risk of bias will be identified across seven domains: sequence generation; allocation concealment; blinding of participants, personnel and outcome assessments; completeness of outcome data; selective reporting of outcomes; other sources of bias. Each domain will be rated as high risk, low risk or unclear risk.

Qualitative data within process evaluations will be appraised using a tool developed in previous systematic reviews.[45 46] It will address two key domains: reliability and trustworthiness (sampling rigour; data collection rigour; analysis rigour; data supporting analysis); and usefulness (breadth/depth of findings; privileges participant perspectives). Two global assessments of overall reliability/trustworthiness and overall usefulness will then be made. Domains are rated as high, medium or low.

Economic data will be appraised with the Drummond checklist.[47] It will assess if there are appropriate descriptions of comparators; identification and valuation of costs and consequences; discounting; and analysis of uncertainty. We will also consider the governance and ethical conduct of studies.[48] All quality appraisal will be undertaken independently and in duplicate, with disagreement being resolved through discussion or recourse to a third reviewer.

### Synthesis of results
#### Programme theory, context, implementation and acceptability (RQ1; RQ4; RQ5)
We will construct a single taxonomy describing intervention types, theories and outcomes. It will be used to understand if there are different or dominant theories according to different types of interventions and/or outcomes.[43] Context and implementation will be classified according to the CICI framework.[42] Implementation will also comprise data on quantifiable assessments of implementation, including: reach; receipt; and fidelity. Acceptability will draw on the understanding, as outlined by Medical Research Council (MRC) guidance on process evaluations.[49] It will include simplistic measures of satisfaction to more complex, often qualitative considerations of the experience of engaging with the intervention. Acceptability will be explored for participants, delivery agents and service funders.

Analysis will adhere to the five phases of framework synthesis[50–53]: (1) familiarisation with included studies; (2) develop a thematic framework based on the parameters of the review. This will be refined and calibrated with a subset of data; (3) index the remaining corpus of data. New themes, and associated codes, may be generated in vivo through the process of constant comparison across studies and against the framework; (4) charting the data into a framework matrix summarising data by category from each transcript; (5) mapping and interpreting data to create the typology of interventions and to explain associations between themes to help understand how and why interventions may or may not be effective. The synthesis may be presented graphically and narratively.

### Intervention outcome effects, equities and economic effects (RQ2)

The taxonomy describing intervention types, theories and outcomes will inform and structure the analyses undertaken as part of the quantitative synthesis. Depending on intervention heterogeneity we will consider meta-analysing effect estimates from outcome evaluations. In preparation for meta-analysis, we will extract effect estimates from studies that will be classified into outcome domains. Where appropriate outcomes will be converted to ORs using logistic transformation for pooling. Estimates from cluster randomised trials will be checked for unit of analysis issues, and where necessary, an inflation factor will be applied to the SE of effect estimates. Where intracluster correlation coefficients (ICC) are not available and effect estimates have not been adjusted for clustering, we will impute an ICC using the average of estimates for specific outcomes from 'most similar' intervention evaluations.

We will undertake robust variance estimation meta-analyses according to intervention type, outcome and timepoint, considering up to 6 months from baseline as short term, 6 months to 2 years as mid term, and beyond 2 years as long term. Robust variance estimation meta-analysis is a method that permits the inclusion of more than one effect estimate per study in a meta-analysis; this is in contrast to standard meta-analysis models that assume independence between individual effect estimates. It is common in meta-analysis of psychosocial interventions for outcome evaluations to present multiple relevant effect estimates per outcome (eg, multiple estimates of child behavioural problems). This method will permit use of all relevant information from included studies. Within each meta-analysis, we will examine heterogeneity using a combination of Cochran's Q, tau-squared and $I^2$. Where heterogeneity is substantial ($I^2$ >50%), we will scrutinise included studies to hypothesise and explore the reasons for this.

Equity effects will be categorised according to PROGRESS-Plus.[54] Data will be reported in adherence to the PRISMA E-2012 extension.[55] We will use harvest plots to assess equity effects and metaregression to test whether characteristics of study populations are associated with effectiveness. Intervention harms will be treated in accordance with the PRISMA harms extension.[56]

Economic evaluations will be summarised. Summarised data will include measures of costs, cost-effectiveness, indirect resource use, and whether a trial-based or model-based analysis was conducted. If there is sufficient homogeneity in measures across studies, these will be synthesised via meta-analysis. Measures of costs, cost-effectiveness and indirect resource use will be adjusted in line with inflation and currency to provide a contextually relevant estimate of costs in the current UK context.

### Synthesis of qualitative and quantitative syntheses

The review will adhere to a convergent synthesis design.[57 58] This will entail the separate synthesis and reporting of qualitative and quantitative data in methodologically relevant but complementary manner, before integrating them in a further synthesis (figure 1). At this stage, the synthesis of context, implementation and acceptability will be integrated with outcome data to explain intervention effectiveness and potential inequities. To support this, we will use a narrative summary and a 2×2 matrix that

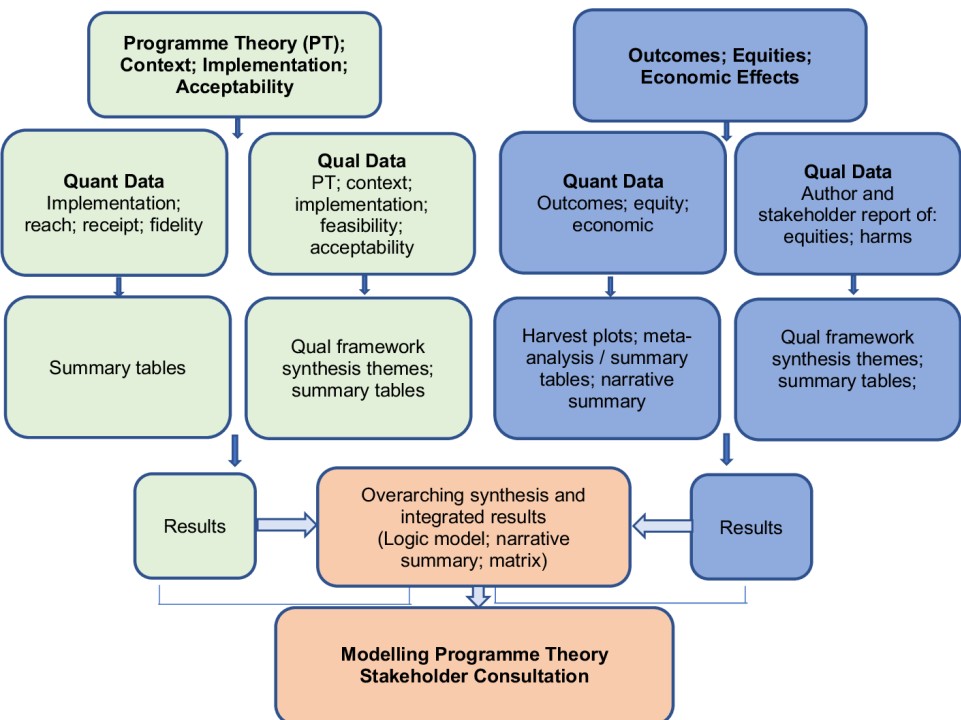

**Figure 1** Children and young people's interventions to improve mental health and well-being outcomes: systematic review results-based convergent synthesis design.

will map context, implementation, acceptability, participant values and costs against the included intervention theories/types.[59] The matrix can demonstrate where an intervention may have (in)sufficiently accommodated factors that led to its (in)effectiveness, and what factors need to be better attended to moving forward. We will additionally develop a logic model(s) of a potentially effective programme theory or theories that is a best fit with the evidence base.

To support harmonisation and integration of syntheses across the review, we have formulated research questions that are complementary and contingent, where achieving a comprehensive answer to one question is dependent on the answers to other questions. When conducting the review, we will: (1) have the same members of the research team work on both the synthesis of qualitative and quantitative data; (2) screen all study types simultaneously and by the same members of the research team, storing data in EPPI Reviewer to ensure easy sharing of study data across syntheses; and (3) use method specific appraisal tools that have been combined in previous reviews due to providing epistemological flexibility or consonance. We will consider appraising the certainty of evidence generated through the different reviews with the compatible tools of GRADE[60] and GRADE CERQual.[61]

### Programme theory modelling, stakeholder consultation and intervention prioritisation

From the review findings, we will aim to identify an overarching programme theory (or compatible programme theories across socioecological domains), and associated interventions, which can address multiple outcomes across a number of populations (RQ5). These will be diagrammatically depicted with the use of system-based logic models and an accompanying narrative summary.[62]

Using the review logic model as a departure point, we will integrate review data to clarify and expand on key intervention domains: theory (theories of change); setting; population; context; implementation; and outcomes. The logic mode (s) then will illustrate the relationship between the underpinning theories of change, intervention effectiveness and important modifying factors that impact on implementation and acceptability.

Stakeholder consultations will be held to prioritise the theories and interventions that might be most feasible and acceptable within the UK context and which may progress to further evaluation (RQ6) (see patient and public involvement section for stakeholder description). There are three phases of assessment as part of this prioritisation process (figure 2):

### Phase 1 intervention identification

Stakeholders will assess and rank candidate programme theories, and associated interventions, against the following progression criteria: (1) acceptability; (2) feasibility (particularly in UK context); and (3) perceived effectiveness. At the end of this phase, there should be a potential programme theory and possible intervention components. Where no candidate theories meet the progression criteria, or an overarching theory is identified but there are no intervention components (eg, no intervention embodies all of the interacting aspects of the theory), we will consider that de novo intervention development is required.

### Phase 2 intervention development and adaptation

If a programme theory is identified, and has an associated intervention, stakeholders will assess if adaptation is required for the UK context (if evaluated elsewhere). This will be done by mapping similarities in contexts using a

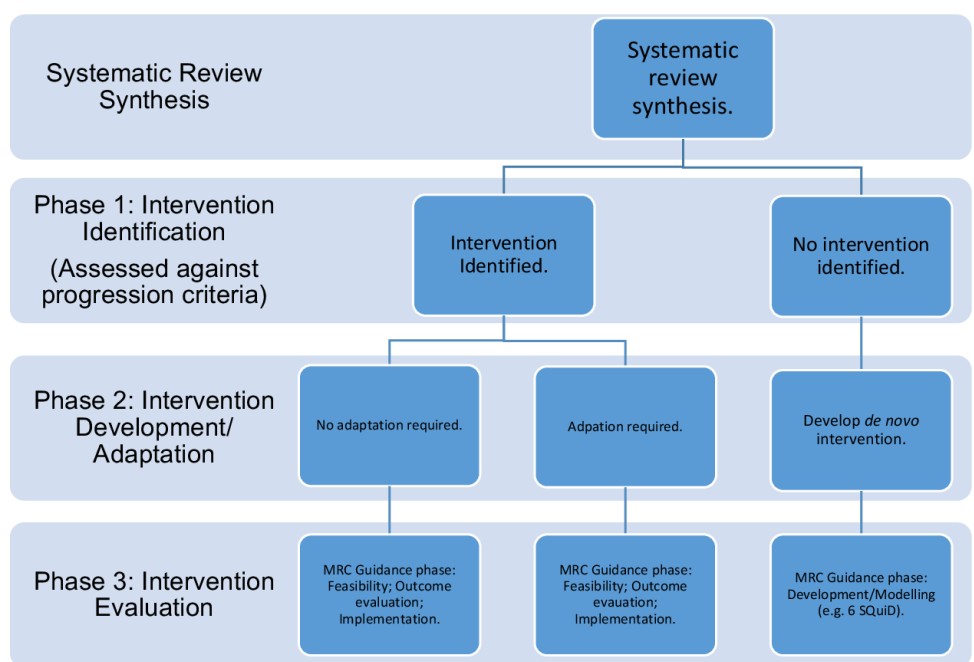

**Figure 2** Intervention prioritisation for development, adaptation and evaluation in UK context.

simplified version of the CICI framework.[42] If contexts are considered dissimilar, stakeholders will consider the types of adaptations that may be required. MRC-NIHR funded guidance on adaptation will support this discussion.[63 64 65] If a de novo intervention is deemed necessary, we will ask stakeholders to undertake preliminary consultation on what an intervention might look like, considering what contextual factors need to be accommodated in order to make it acceptable, feasible and effective.

### Phase 3 intervention evaluation

This phase will consider the appropriate research design for progressing the intervention. We will use the MRC's guidance on developing and evaluating complex interventions to define the phase of evaluation.[66] To support decision-making in the case where it is appropriate to evaluate an existing intervention (adapted or non-adapted), we will use the 'borrowing strength' framework to assess if any outcome data from the original context is applicable within the UK context.[29] The framework dictates that if contexts are largely congruent or adaptations are minimal, then an implementation study may be warranted. Where contexts are significantly dissimilar or substantial adaptation has been undertaken, more extensive feasibility and outcome evaluation will be required. It is likely that decision-making at this phase will be predominantly conducted by the research team. On completion of the three prioritisation phases, we anticipate having a clearly identified research agenda to progress beyond the review.

## ETHICS AND DISSEMINATION
### Ethics

Cardiff University's School of Social Sciences Research Ethics Committee will provide ethical approval for the stakeholder consultation if required.

### Patient and public involvement

The review includes a comprehensive programme of PPI. Engagement with children and young people will be facilitated through Children's Social Care Research and Development Centre (CASCADE) Voices, a collaboration between Voices from Care Cymru and the CASCADE at Cardiff University. CASCADE Voices comprises a research advisory group of care-experienced young people. We will consult with them at study commencement to refine and confirm the scope and focus of the review, notably the key subgroups to be analysed and the contextual influences on implementation.

To support the prioritisation of interventions moving forward, we will hold afurther consultation with CASCADE Voices. A social care practitioner workshop will be hosted with the ExChange Wales Network. The network, hosted by CASCADE, brings together social care sector stakeholders to share experiences and expertise. Further consultation opportunities will be sought with the UKRI funded Transdisciplinary Research for the Improvement of Young Mental Public Health (TRIUMPH) Network. The CHIMES study principal investigator (RE) coleads the networks' programme on the mental health of key groups, which includes care-experienced young people.

### Dissemination

Outputs from the study will be (1) NIHR-PHR monograph; (2) academic publications reporting the results; and (3) a briefing report for policy and practitioners. Presentations will be delivered at academic, policy and practice workshops and conferences, both nationally and internationally. Networks and research infrastructures will be utilised to disseminate findings, notably TRIUMPH and the What Works Centre for Children's Social Care. The latter organisation was funded by the Department for Education and supported by Cardiff University, with the aim of improving outcomes for children in care. It has a focus on evidence synthesis, providing a depository of evidence summaries on their website for social care professionals. Further dissemination activities will be identified in collaboration with PPI stakeholders throughout the course of the study. Moving forward, sharing of knowledge on which interventions are likely to work means that finite public resources can be expended on approaches that can best ensure the positive mental health and well-being of care-experienced children and young people, while encouraging disinvestment from ineffective and potentially harmful approaches.

**Twitter** Rhiannon Evans @1RhiannonEvans, Maria Boffey fosteringnet; @tfn_Wales, Jane Noyes @janenoyes, Simone Willis @WillisWellbeing and Charlotte Wooders @ fosteringnet, tfn_Wales

**Contributors** RE secured funding for the study. RE, SM, HEM, JN, MR, GJM-T, RT and SW contributed to the design of the systematic review. SW and HEM developed the search strategy. MB and CW advised on the PPI engagement and stakeholder consultation strategy. REE drafted the protocol. All authors read and approved the final manuscript. RE is the guarantor of the review.

**Funding** This work was supported by the NIHR-PHR grant number NIHR129113. *This work was supported by The (DECIPHer) funded by Welsh Government through Health and Care Research Wales. Centre for Development, Evaluation, Complexity and Implementation in Public Health Improvement*. The Centre for Trials Research receives funding from Health and Care Research Wales and Cancer Research UK.

**Competing interests** None declared.

**Patient consent for publication** Not required.

**Provenance and peer review** Not commissioned; externally peer reviewed.

**ORCID iDs**
Rhiannon Evans http://orcid.org/0000-0002-0239-6331
Jane Noyes http://orcid.org/0000-0003-4238-5984
Helen E Morgan http://orcid.org/0000-0003-2470-9746
Rob Trubey http://orcid.org/0000-0002-9550-1785
Michael Robling http://orcid.org/0000-0002-1004-036X
Simone Willis http://orcid.org/0000-0003-3949-7651

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
