## [Reviewer comments · BMJ Open]

ARTICLE DETAILS

TITLE (PROVISIONAL)	Care-experienced children and young people's Interventions to improve Mental health and Wellbeing outcomes: Systematic review (CHIMES) Protocol
AUTHORS	Evans, Rhiannon; Boffey, Maria; MacDonald, Sarah; Noyes, Jane; Melendez-Torres, G.J; Morgan, Helen; Trubey, Robert; Robling, Michael; Willis, Simone; Wooders, Charlotte

VERSION 1 – REVIEW

REVIEWER	Emma Sonesson University of Cambridge, United Kingdom
REVIEW RETURNED	03-Sep-2020

GENERAL COMMENTS	Many thanks for the opportunity to review this protocol. The planned systematic review (and potential meta-analysis) focuses on reviewing the evidence for interventions to improve the mental health of and wellbeing of care-experienced children. The authors have carefully planned this analysis such that it builds upon prior reviews and fills important gaps in the literature, particularly as regards context and implementation. All analyses, both qualitative and quantitative, have been carefully planned and are described in detail in the protocol. It is further clear that the authors place value in involving key stakeholders in the design and interpretation of the review. Major comments 1) My one main concern is the inclusion of Research Question 6 (i.e. What do stakeholders think is the most effective, feasible and acceptable intervention in the UK that could progress to further outcome or implementation evaluation?) within this review. Without a doubt, this is a relevant and important question which will need to be addressed as the authors move forward in their empirical work. However, I am not convinced that it is an appropriate question for a systematic review, which in general should try to synthesise evidence in absence from the type of personal judgements and prioritisation presented in their stakeholder consultation plan. I would strongly recommend that this question be included in a separate publication in order to maintain clear boundaries of the systematic review and any further intervention design/evaluation. Minor comments 1) It would be helpful if the authors could provide a more detailed description of Research Question 5 and the methods specific to addressing this question. 2) A couple of clarifications for the inclusion/exclusion criteria: a. How will the authors deal with studies that do not fit neatly inside
--

	of the 0-25 boundary age range? For example, could a study be included if it focuses on young adults aged 18-30? b. When the authors say they will exclude studies from 'lower-middle income' countries, do they mean low-and-middle-income? If not, they should justify this criterion as it seems quite specific. c. How would the authors treat combined interventions that include both psychosocial AND pharmacological components? And, are there any restrictions on the proportions of participants receiving pharmacological treatment separate from the intervention itself? d. How are the authors defining the 'acceptability' outcome? They state that 'acceptability will be considered as the experiences of all stakeholders, including participants, delivery agents and service funders, and their interactions with the interventions', but this is quite broad and could benefit from additional description (it is clearer how they are defining implementation and context). 3) It would be helpful if the authors could clarify exactly which tools they will use for quality appraisal - this section is not detailed enough, in my opinion. Best wishes to the authors as they complete this important review.
--	---

REVIEWER	Matthew Hamilton Orygen
REVIEW RETURNED	05-Sep-2020

GENERAL COMMENTS	A high quality protocol for research to address an important topic. As this work explores economic topics and would no doubt be useful background for future economic models, authors might consider explicitly capturing health utility measures reported in the evidence reviewed.
--

REVIEWER	Thomas King University of Oxford, UK
REVIEW RETURNED	10-Sep-2020

GENERAL COMMENTS	This protocol gives a detailed and structured overview of this ambitious review, beginning with a concise research background that leads logically on to a clear set of aims. The intended methodology is clear and thorough, with a large and broad range of databases to be searched and a detailed example search provided. It is also promising to see the inclusion of relevant stakeholders in the review process. This review would appear to be a valuable contribution to the current evidence base.
---

VERSION 1 – AUTHOR RESPONSE

Reviewer 1	
The planned systematic review (and potential meta-analysis) focuses on reviewing the evidence for interventions to improve the mental health of and wellbeing of care-experienced children. The authors have carefully planned this analysis such that it builds upon prior reviews and fills important gaps in the literature, particularly as regards	We would like to thank the reviewer for their comments in support of the manuscript.

context and implementation. All analyses, both qualitative and quantitative, have been carefully planned and are described in detail in the protocol. It is further clear that the authors place value in involving key stakeholders in the design and interpretation of the review.	
My one main concern is the inclusion of Research Question 6 (i.e. What do stakeholders think is the most effective, feasible and acceptable intervention in the UK that could progress to further outcome or implementation evaluation?) within this review. Without a doubt, this is a relevant and important question which will need to be addressed as the authors move forward in their empirical work. However, I am not convinced that it is an appropriate question for a systematic review, which in general should try to synthesise evidence in absence from the type of personal judgements and prioritisation presented in their stakeholder consultation plan. I would strongly recommend that this question be included in a separate publication in order to maintain clear boundaries of the systematic review and any further intervention design/evaluation.	We appreciate this comment and recognise the potential value of separating out RQ6. However, we do not feel that we are in a position to completely remove this question, as we have been funded to address all RQs by the NIHR-PHR panel. As such, we feel we should present the complete work we propose to undertake. We also note that it is considered a marker of good practice to have key stakeholder engagement in systematic reviews. Their role is to contribute to all aspects of the review design and conduct either as a co-author or in an advisory capacity. It is universally accepted that the outcomes of interest to patients and service recipients are often different to those of professionals and reviewers. It is therefore vital that systematic reviews explore these differences to ensure the applicability of the findings to real life. Cochrane has developed a consumer network for this purpose and there is a chapter in the Cochrane handbook about how to involve consumers in all aspects of review design and conduct. We have followed this guidance and feel that it will enable us to conduct a more robust and applicable review. However, we have aimed to create some clearer separation between RQ1-5 and RQ6 within the 'Review Aims and Questions' section (p 5, In 32-32 (Track-changed version)). I
It would be helpful if the authors could provide a more detailed description of Research Question 5 and the methods specific to addressing this question.	We have included more detail on how we will answer RQ5 by integrating review detail into the review logic model (p 12, In 11-16).
Inclusion/exclusion criteria: How will the authors deal with studies that do not fit neatly inside of the 0-25 boundary age range? For example, could a study be included if it focuses on young adults aged 18-30?	We have specified in the inclusion criteria that we will include studies with a focus on young adults (e.g. 18-30 years) only if subgroup data is provided for participants aged under 25 years old (p 6 , In 15-17).
Inclusion/exclusion criteria: When the authors say they will exclude studies from 'lower-middle income' countries, do they mean low-and-middle-income? If not, they should justify this criterion as it seems quite specific.	We do mean low- and middle-income countries and have corrected this in the inclusion criteria.
Inclusion/exclusion criteria: How would the authors treat combined interventions that include both psychosocial AND pharmacological components? And, are there	We have specified that we will not include pharmacological interventions if they are single component or combined with psychosocial components (p 6, In 31-32). We have also

any restrictions on the proportions of participants receiving pharmacological treatment separate from the intervention itself?	stated that there are no restrictions of the pharmacological treatment being received separate from the intervention (p 6, ln 20-21).
Inclusion/exclusion criteria: How are the authors defining the 'acceptability' outcome? They state that 'acceptability will be considered as the experiences of all stakeholders, including participants, delivery agents and service funders, and their interactions with the interventions', but this is quite broad and could benefit from additional description (it is clearer how they are defining implementation and context).	We understand that we have taken a broad definition of acceptability, in part because we would like to theorise from the review how acceptability is operationalised. We have given a clearer statement on what is included in our definition, in accordance with MRC guidance on process evaluations (Moore et al., 2014). We have referenced the guidance to indicate the source of our understanding (p 10, ln 7-11).
It would be helpful if the authors could clarify exactly which tools they will use for quality appraisal - this section is not detailed enough, in my opinion.	We have added detail on the quality appraisal tools used. We had originally provided a brief description due to word count restrictions (p 9, ln 13-34).
Reviewer 2	
A high quality protocol for research to address an important topic.	We would like to thank the reviewer for their comments in support of the manuscript.
As this work explores economic topics and would no doubt be useful background for future economic models, authors might consider explicitly capturing health utility measures reported in the evidence reviewed.	We appreciate this suggestion and have stated that we will extract health utility measures (p 9, ln 8-9).
Reviewer 3	
This protocol gives a detailed and structured overview of this ambitious review, beginning with a concise research background that leads logically on to a clear set of aims. The intended methodology is clear and thorough, with a large and broad range of databases to be searched and a detailed example search provided. It is also promising to see the inclusion of relevant stakeholders in the review process. This review would appear to be a valuable contribution to the current evidence base.	We would like to thank the reviewer for their comments in support of the manuscript.
Formatting Style	
The format for Protocol papers in BMJ Open does not include a conclusion section so please remove that section and incorporate any important details in another part of the manuscript.	We have removed the conclusion section of the manuscript and integrated the key points into an early section (p 14, ln 11-15).
Figure/s should not be embedded Please remove figure/s from main document and upload each of them separately under file designation 'Image' (except tables and please ensure that figures are in better quality or not pixelated when zoomed in).	We have removed the figures from the main document and uploaded them as separate files.

VERSION 2 – REVIEW

REVIEWER	Emma Soneson
-----------------	--------------

	University of Cambridge, UK
REVIEW RETURNED	16-Nov-2020

GENERAL COMMENTS	I am pleased with the way the authors have revised the manuscript and believe it is ready for publication. I particularly appreciate their detailed response to my main concern (i.e. the inclusion of RQ6) - I think the relatively simple separation they have made between questions addresses this concern in a way that maintains objectivity in the review whilst still highlighting the importance of advisory groups in the review process. I look forward to seeing the result of this protocol. All the best to the authors as they complete the review.
--